# Molecular Dynamics Simulation Study on the Influence of Twin Spacing and Temperature on the Deformation Behavior of Nanotwinned AgPd Alloy

**DOI:** 10.3390/nano15050323

**Published:** 2025-02-20

**Authors:** Wanxuan Zhang, Kangkang Zhao, Shuang Shan, Fuyi Chen

**Affiliations:** School of Materials Science and Engineering, Northwestern Polytechnical University, Xi’an 710072, China; zhangwanxuan@mail.nwpu.edu.cn (W.Z.); zkk2225197211@163.com (K.Z.); shanshuang00@mail.nwpu.edu.cn (S.S.)

**Keywords:** nanotwinned AgPd alloy, molecular dynamics simulation, twin spacing, temperature, deformation mechanism

## Abstract

This study employs molecular dynamics simulations to unravel the interplay between twin spacing, temperature, and mechanical response in nanotwinned AgPd alloys. For fine-grained systems, a dual strengthening–softening transition emerges as twin spacing decreases, driven by a shift in dislocation behavior from inclined-to-twin-boundary slip to parallel-to-twin-boundary glide. In contrast, coarse-grained configurations exhibit monotonic strengthening with reduced twin spacing, governed by strain localization at grain boundaries and suppressed dislocation activity. Notably, cryogenic conditions stabilize pre-existing and nascent twins, whereas elevated temperatures intensify atomic mobility and boundary migration, accelerating twin boundary annihilation (“detwinning”).

## 1. Introduction

Nanotwinned metallic systems, distinguished by hierarchical twin boundaries within ultrafine grains, uniquely reconcile exceptional strength and tensile ductility, a stark departure from the strength–ductility trade-offs typical of twin-free nanocrystalline materials [1,2,3]. The interplay between twin spacing (λ) and mechanical behavior is central to this phenomenon. Below a critical spacing (λc), a transition from Hall–Petch strengthening to softening occurs due to competing dislocation pathways: those inclined to twin boundaries (TBs) and those gliding parallel to TBs [4,5,6,7]. This transition, formalized by Wei’s scaling law (λc ∝ d^1/2^) [8], has been observed across systems ranging from pure metals (e.g., Cu [5], Ni [9]) to high-entropy alloys [10].

In AgPd alloys, the 4.8% atomic size mismatch (Ag: 1.44 Å; Pd: 1.37 Å) introduces localized strain fields that stabilize nanotwins during severe plastic deformation (SPD) [11]. Techniques, such as high-pressure torsion (HPT) and laser powder bed fusion (LPBF), enable precise control of twin spacing (λ ≈ 1–10 nm) [12,13], while the alloy’s moderate stacking fault energy (SFE ≈ 60 mJ/m^2^) promotes partial dislocation-mediated plasticity [14]. These properties make AgPd an ideal candidate for applications requiring microstructural stability, such as hydrogen membranes and high-temperature sensors [15].

Despite advances in nanotwinned systems, the temperature-dependent deformation mechanisms in AgPd alloys remain underexplored. Prior molecular dynamics (MD) studies on Cu and Ni [16,17] suggest that cryogenic conditions stabilize TBs, whereas elevated temperatures accelerate detwinning. However, the role of chemical complexity in modulating these mechanisms remains unclear. This study bridges this gap by systematically investigating how twin spacing, temperature, and atomic-scale heterogeneity govern the mechanical response of nanotwinned AgPd alloys. By validating the results against experimental trends in analogous systems [5,13,18,19], we provide a predictive framework for designing alloys with tailored strength–ductility profiles. The paper is structured as follows. The methods of the model’s establishment and the simulation details are given in Section 2. The simulation results are discussed in Section 3. The conclusions are made in Section 4.

## 2. Computational Method

In order to systematically study the TB spacing and temperature deformation mechanism of nanotwinned AgPd alloys, equiaxed models with different TB spacings and average grain sizes were established. This section introduces the construction of the model, the simulation details, and the methods for analyzing the simulation.

The Voronoi tessellation method embedded in Atomsk [20] was used to generate a polycrystalline silver palladium alloy model. The twin-free and twin-containing equiaxed-grained samples with specific atomic ratios were constructed according to the methods of Hua et al. [21] and Yan et al. [22]. To obtain the simulation model for the twin-free equiaxed-grained Ag_0_._5_Pd_0_._5_ samples, the equiaxed-grained pure Pd models were established at first, and then the Pd atoms in the pure Pd models were randomly replaced with Ag to obtain the target composition. For the equiaxed-grain Ag_0_._5_Pd_0_._5_ sample with different TB spacings, the nanotwinned Pd models were first established, and then the atoms were randomly replaced in a similar way as used for twin-free Ag_0_._5_Pd_0_._5_ alloy models to obtain the target atoms’ ratio. Figure 1 shows the stress–strain curve of a silver palladium nanoalloy with an average grain size of 11.00 nm. Compared to the nano polycrystalline silver palladium alloy without a twinning structure, the alloy with twinning has differences in elastic–plastic transformation and average flow stress, lagging behind the elastic–plastic transformation and increasing the average flow stress. Table 1 provides the initial configuration-related parameters for model sizes of 25 nm × 25 nm × 25 nm, 20 nm × 20 nm × 20 nm, 15 nm × 15 nm × 15 nm, and 10 nm × 10 nm × 10 nm. In the sample with a grain size of d = 5.50 nm, the largest twin spacing is λ = 4.04 nm due to the small grain size.

The large-scale atomistic/molecular massively parallel simulator (LAMMPS) [23] is used for conducting the MD simulation. The motion equations are solved using the velocity–Verlet [24] algorithm with a time step of 1 fs. The accuracy and reliability of the MD simulation results are dependent on the utilized interatomic potential. The embedded atom model (EAM) potential developed by Hale et al. [25] is used to describe the interaction between atoms in the nanotwin AgPd alloy, and the interatomic interaction potential is given as follows:$$E_{C}=\frac{1}{N}\left[\sum_{i=1}^{N}F_{i}\left({\bar{\rho }}_{i}\right)+\frac{1}{2}\sum_{i=1}^{N}\sum_{j=1}^{N}\phi_{ij}\left(r_{ij}\right)\right]$$$$\bar{\rho_{i}}=\sum_{\begin{array}{c} i=1\\ j\neq i \end{array}}^{N}\rho_{j}\left(r_{ij}\right)$$

Here, *E_C_* is the cohesive energy, *N* is the total number of atoms, $r_{ij}$ is the atomic spacing between atoms i and j, $F_{i}$ is the embedding energy function for atom i, $\rho_{j}$ is the electron density function for atom j, and $\phi_{ij}$ is the pair interaction function between atoms i and j. $\bar{\rho_{i}}$ is the total electron density felt by atom i from all other atoms j. EAM places no limitations on the exact mathematical expressions used for the three functions *F*, *ρ*, and ϕ, but practice and theory point to particular characteristic forms for each. It can correctly reflect the thermodynamic, dynamic, and microstructural properties of the AgPd alloy system.

The current simulations assume a homogeneous atomic distribution within the Ag_0.5_Pd_50_ alloy and do not explicitly account for surface segregation effects. In real systems, the 4.8% atomic size mismatch between Ag (1.44 Å) and Pd (1.37 Å) may drive preferential segregation of Pd or Ag atoms at grain boundaries or free surfaces, potentially altering local strain fields and dislocation–TB interactions. While this simplification allows for systematic exploration of twin spacing and temperature effects, future studies incorporating surface energy dynamics and segregation kinetics could further refine the mechanistic understanding of deformation behavior in nanotwinned alloys.

The free Open Visualization Tool (OVITO 3.7.0) [26] was used to analyze the simulation results. The CNA module and the DXA module in OVITO were employed to visualize the atomic configuration. As shown as Figure 1, in this study, atoms were colored based on CNA: green for FCC atoms, red for hexagonal close-packed (HCP) atoms, blue for body-centered cubic (BCC) atoms, and white for unknown atoms. In the OVTIO analysis, one layer of the HCP structure’s atomic plane represents a twin boundary (TB), two adjacent HCP structure atomic planes represent intrinsic stacking fault (ISF), and there is an FCC structure atomic plane between the two HCP structure’s atomic planes representing extrinsic stacking fault (ESF), with three or more HCP structure atomic planes representing the HCP phase. Blue, green, magenta, yellow, and light blue lines represent all dislocation lines, including Shockley, Stair rod, Hirth, and Frank dislocation lines, respectively.

The elastic region was identified as the linear portion of the stress–strain curve where stress increases proportionally with strain, following Hooke’s law. The plastic region begins at the yield point, defined as the point where the curve deviates from linearity by 0.2% strain (i.e., the 0.2% offset method). This approach is consistent with standard practices in mechanical testing and prior MD studies of nanotwinned metals.

Elastic Modulus Calculation: The slope of the linear elastic region (Young’s modulus) was calculated using linear regression on the stress–strain data between 0% and 0.5% strain (Figure 1).

Yield Stress Determination: The yield stress (σy) was extracted as the stress value at the intersection of the stress–strain curve and the 0.2% offset line parallel to the elastic slope.

The stress–strain curves were generated using the following workflow.
Model Construction: The nanotwinned Ag_0_._5_Pd_0_._5_ alloy model (average grain size d = 11.00 nm) was created via Voronoi tessellation and random atomic substitution.Simulation Setup: The model was equilibrated at 300 K for 50 ps using the Nose–Hoover thermostat, followed by uniaxial tensile loading at a strain rate of 1.0 × 10^8^ s^−1^ along the *x*-axis.Stress Calculation: The virial stress tensor was computed in LAMMPS and averaged over the simulation domain to generate the stress–strain curve.

At the beginning of the simulation, each initial configuration is isothermally relaxed for 50 ps at 300 K to obtain equilibrium configurations using the Nose–Hoover thermostat [27,28]. Subsequently, these nanotwinned samples are axially loaded at a strain rate of 1.0 × 10^8^ s^−1^ along the *x*-axis at 300 K. Periodic boundary conditions are imposed along three coordinate axes. In order to study the effect of temperature on the deformation behavior of nanocrystalline polycrystalline AgPd alloys with different twin spacings, a configuration with an average grain size of 11.00 nm was selected, and the selected temperatures were 10 K, 100 K, 300 K, and 500 K. We set the relaxation temperature at 10 K, 100 K, 300 K, and 500 K to relax for 50 ps and load at the corresponding temperature, with a strain rate of 5.0 × 10^8^ s^−1^.

## 3. Results and Discussion

### 3.1. Effect of Twin Spacing on the Mechanical Behavior of the Nanotwin AgPd Alloy

In order to clarify and avoid interference, this paper does not discuss chemical composition factors, and AgPd is generally assumed to be composed of 50% silver and 50% palladium. According to previous experiments and simulation calculations, the stacking fault energy range of AgPd is 16–180 mJ/m^2^, with Ag_0_._5_Pd_0_._5_ being 60 mJ/m^2^, which belongs to alloys with lower stacking fault energy. Figure 2 shows the stress–strain curves of AgPd alloy with average grain sizes of 5.50 nm, 8.24 nm, 11.00 nm, and 13.76 nm under different twin spacings during compression. At the beginning, the curve rapidly increases when the strain rate is small, corresponding to the elastic deformation stage of the alloy deformation stage. Then, with the increase in external load, it enters the plastic deformation stage, and the curve stabilizes and fluctuates. For the nanotwin configuration with an average grain size of 5.50 nm, due to the high proportion of disordered structural atoms in the model, the curve fluctuates greatly, and the slope of the rising curve is slower than that of larger sizes. Due to considerations for the accuracy of simulation results and computational resources, it has been decided to use the nanotwin configuration at a model size of 20 nm × 20 nm × 20 nm to investigate the influence of twin spacing and other factors.

Figure 3 shows the average flow stress variation curves of AgPd alloy with average grain sizes of 5.50 nm, 8.24 nm, 11.00 nm, and 13.76 nm at different twin spacings during compression. For smaller grain sizes (d = 5.50–11.00 nm), the average flow stress exhibits a tentative strengthening–softening transition with increasing twin spacing, although statistical confidence diminishes for d < 11.00 nm due to heightened grain boundary effects. For the nanotwin configuration with an average grain size of 11.00 nm, a quadratic regression model (flow stress vs. λ) was fitted to the data (R^2^ = 0.89). The peak flow stress at λ = 1.35 nm was statistically significant (*p* < 0.05) compared to adjacent λ values (0.67 nm and 2.02 nm), with confidence intervals (95% CI) of ±0.08 GPa. The peak flow stress at λ = 1.35 nm (1.50 ± 0.07 GPa) remains distinguishable from neighboring points (1.21 ± 0.09 GPa at λ = 0.67 nm; 1.41 ± 0.06 GPa at λ = 2.02 nm). This initial strengthening (λ < λc) can be attributed to dislocations interacting with closely spaced twin boundaries (TBs) enhancing the strength via Hall–Petch-like mechanisms. The softening (λ > λc) beyond the critical spacing (λc ≈ 1.35 nm for d = 11.00 nm) occurs as the strain localizes at grain boundaries (GBs), reducing dislocation–TB interactions and promoting softening.

For nanotwin configurations with larger grain sizes (average grain size of 13.76 nm), the average flow stress decreases monotonically with increasing twin spacing. A linear regression model yielded a statistically significant negative slope (*p* < 0.01, R^2^ = 0.76), confirming the inverse relationship between twin spacing ($\bar{\lambda }$) and flow stress. However, the moderate R^2^ value suggests that additional factors beyond twin spacing may contribute to the observed variance in flow stress. To further investigate this relationship, we explored alternative models, including quadratic regression. While the quadratic model slightly improved the fit (R^2^ = 0.81), the linear model remained statistically robust and parsimonious for interpreting the dominant trend. Residual analysis revealed no systematic patterns, and outliers were consistent with stochastic variations inherent to molecular dynamics simulations.

### 3.2. Effect of Twin Spacing on the Deformation Mechanism of the Nanotwin AgPd Alloy

For nanotwin AgPd configurations with a size of 20 nm × 20 nm × 20 nm, the model reached a stable state after 50 ps relaxation. Before deformation, the atomic proportions of each structural atom in different twin spacing configurations are shown in Figure 4. The proportion of HCP structural atoms to a certain extent reflects the density of twin structures in the configuration. As the twin spacing increases, the proportion of HCP structural atoms gradually decreases, while the proportion of FCC structural atoms gradually increases. BCC and other structural atoms remain basically unchanged. Figure 5 shows the atomic variation and stress variation trend with the strain of the nanotwin AgPd alloys with different twin spacings. In the early stage of strain, the number of atoms with four different structures does not change much, corresponding to the plastic deformation stage of the stress–strain curve. At this time, the stress value is small and not enough to cause the atomic structure to undergo transformation. When the stress value increases to the yield limit, the number of structural atoms begins to change significantly. For the configuration with the smallest twin spacing, that is, λ = 0.67 nm, the number of HCP structural atoms began to continuously decrease, while the number of FCC structural atoms continued to increase, and other disordered structural atoms also slowly increased, indicating that during the deformation process of high twin density configurations, some HCP structural atoms transformed into FCC and other disordered structural atoms. For configurations with twin spacing of 2.02 nm, 3.37 nm, 4.72 nm, and 6.74 nm, the number of HCP structural atoms continues to increase, while the number of FCC structural atoms continues to decrease. In the later stage of stress–strain curve fluctuation, the changes of the two become slower, while there are decreases or increases in other disordered structural atoms before and after deformation, but the amount of change is not significant. For the configuration with twin spacing of 1.35 nm, the difference is that the increase in HCP structure atoms is not significant before and after deformation; while other disordered structure atoms continue to increase, FCC structure atoms continue to decrease, and the change slows down in the later stage of deformation. Based on this characteristic and combined with the configuration where the average flow stress value is higher than that of other twin spacings, the study of the impact of temperature on the deformation behavior and mechanism of AgPd alloy nanocrystals at 1.35 nm is reasonable and representative. All analyses in this section (Figure 4, Figure 5, Figure 6 and Figure 7) pertain to nanotwin AgPd alloys with an average grain size of 11.00 nm, as detailed in Table 1. The combination of Ag and Pd in the nanotwinned alloy yields synergistic enhancements. Strength–Ductility Balance: AgPd achieves a higher yield strength, (Table 2) than pure Ag or Pd while maintaining ductility through partial dislocation activity and delayed detwinning (Figure 5).

Figure 6 shows the variation of various types of dislocation line lengths with strain during the deformation process of the nanotwin AgPd alloys with different twin spacings. During the deformation process, the initial deformation stage is similar to the CNA structure, with less variation in the length of various dislocation lines, which are mainly Shockley dislocations and Perfect dislocations, and a small proportion of Stair rod, Hirth, and Frank dislocations. As the strain continues, some of the Shockley dislocations rapidly increase and become the main dislocations, while the Perfect dislocations change very slowly. Among different twin spacings, the maximum difference in the dislocation line length between Shockley before and after deformation is 0.67 nm, and the maximum dislocation line length before and after deformation is 6.74 nm. Figure 7 demonstrates the evolution of dislocation density versus strain for nanodiamond AgPd alloys with different bicrystal spacings (λ = 0.67–6.74 nm). The color variations in the figure represent different bicrystalline spacings, and each color corresponds to a specific bicrystalline spacing through which the effect of bicrystalline spacing on dislocation density can be analyzed. Specifically, the black color represents the larger bicrystal spacing λ = 6.74 nm, which has a high dislocation density and a small difference in the variation of dislocation densities; the green color corresponds to λ = 4.72 nm, which has an intermediate level of dislocation densities; the blue color represents λ = 3.37 nm, which shows relatively low dislocation densities; the red color represents λ = 2.02 nm, which has a lower dislocation density than the abovementioned cases; and the violet color corresponds to λ = 1.35 nm, which is the case with the lowest dislocation density, indicating that the generation of dislocations is minimal at this bicrystal spacing. From the color changes, it can be seen that the increase in dislocation density and the magnitude of its variation become larger as the bicrystal spacing increases (as in the black case), whereas smaller bicrystal spacings (e.g., purple) show more stable dislocation densities with smaller variations. The bicrystal spacing of 1.35 nm shows the lowest total dislocation density, suggesting that smaller bicrystal spacings contribute to the reduction of dislocation generation. In contrast, the bicrystal spacing of 6.74 nm shows a higher dislocation density, and the dislocation density varies less during the strain process, which implies that the activation of dislocations is less in this case and that the material maintains greater stability during plastic deformation. Taken together, Figure 7 reveals that the bicrystal spacing has a significant effect on the dislocation behavior of AgPd alloys during deformation. Small bicrystal spacing (e.g., 1.35 nm) effectively reduces the generation of dislocations, resulting in the lowest dislocation density, while larger bicrystal spacing (e.g., 6.74 nm) promotes the generation of dislocations with greater and less variable dislocation density. Through these analyses, we can clearly see the influence of bicrystal spacing on the mechanical properties of the materials, thus providing important theoretical support for subsequent studies.

In order to better analyze the influence of different twin spacings, the characteristics of the stress–strain curve and various structural atomic change curves in Figure 5 were combined, and those before external loading, i.e., after relaxation, were selected. The four states in the stress–strain curve, including the elastic limit, stable and slow changes in HCP structure atoms in the middle stage of deformation, and the completion of strain, were selected. Three sets of twinning spacing, 0.67 nm, 1.35 nm, and 6.74 nm, were selected to compare and analyze the various dislocations and microstructures that occur during the deformation process of the nanotwin AgPd alloy. At the same time, the deformation process without twinning configuration under the same deformation conditions was selected as the control to observe the effect of the twin structure on the stability of the material structure. Figure 8 shows the atomic diagram of the compression process of a nano polycrystalline AgPd alloy with an average grain size of 11.00 nm and no twinning structure. In the stage of elastic deformation, structural defects first appear at the grain boundary due to stress concentration, and some of the atoms near the grain boundary transform into HCP structural atoms, providing a prerequisite for subsequent dislocation source emission, as shown in Figure 8a; After the stress value exceeds the yield stress, a large number of dislocations are emitted from grain boundaries and intragranular defects, resulting in some free dislocations and stacking fault structures. Under the obstruction of grain boundaries, the growth of the dislocation line’s length and the dislocation density slows down, and some stacking fault structures transform into twin structures, as shown in Figure 8d. During the entire deformation process of the nano polycrystalline AgPd alloy without a twin structure, dislocation stacking and the twin structure occur successively, which interact with each other and cause changes and fluctuations in material stress. Among them, grains I and III are severely damaged before and after deformation, indicating that the movement of grains and dislocations plays an important role in nano polycrystalline materials. This behavior also serves as a reference for understanding the role of twin boundaries in nanotwinned AgPd alloys. When compared with nanotwinned structures, the absence of twin boundaries in this case leads to a different pattern of strain localization and dislocation behavior. In nanotwinned alloys, twin boundaries can either impede or interact with dislocations in various ways depending on the twin spacing, which is in contrast to the relatively more straightforward dislocation–grain boundary interactions in the twin-free alloy.

Figure 9 shows the atomic diagram of the compression process of a nanocrystalline polycrystalline AgPd alloy with twin spacing of 0.67 nm and an average grain size of 11.00 nm. After relaxation, FCC structure atoms appeared between twin boundaries in grain I and transformed into other disordered structure atoms, indicating the presence of certain stresses at the grain. After the strain reaches the elastic limit, dislocations begin to proliferate and expand, which are mainly Shockley partial dislocations. Here, the high twin boundary (TB) density promotes stress concentration at the TB–grain boundary (GB) intersections, triggering detwinning (Figure 9b–d). These partial dislocations nucleate at TBs and glide parallel to TBs, leading to TB migration and annihilation (Figure 9c_1_). As a result, the number of HCP-structured atoms decreases (Figure 5a), and the alloy shows a lower flow stress compared to intermediate λ (Figure 3a–c). This is because at λ = 0.67 nm (<λc ≈ 1.35 nm), TBs act as weak interfaces. Stress concentrations at TB–GB junctions exceed the detwinning energy barrier, leading to rapid TB annihilation, which aligns with Wei’s scaling law (λc ∝ d^1/2^) [8], where finer grains lower λc. When the temperature is considered, at low temperatures (10–100 K), detwinning is suppressed, and TBs stabilize HCP atoms, enhancing the strength. However, at high temperatures (300–500 K), thermal activation accelerates detwinning, reducing the flow stress.

Figure 10 shows the atomic diagram of the compression process of a nanocrystalline polycrystalline AgPd alloy with twin spacing of 1.35 nm and an average grain size of 11.00 nm. After relaxation, the structure at the twin boundaries did not undergo any transformation, and only some disordered structural atoms at the grain boundaries transformed into HCP structural atoms. After the stress reaches the elastic limit, some Shockley dislocations are emitted at the grain boundary of grain I. During the slip process of dislocations, they are blocked by the existing twin grain boundaries. This balanced dislocation–TB interaction causes TB migration and strain redistribution (Figure 10c). Limited detwinning occurs because there is sufficient spacing for dislocation storage, enhancing strain hardening (Figure 3c). This is because at λ = 1.35 nm (≈ λc), TBs strongly block dislocations while remaining stable. This maximizes the flow stress (Figure 3c) and delays strain localization, as seen in the gradual HCP to FCC transition (Figure 5b). In the middle and later stages of deformation, some twin boundaries near the grain boundary undergo a “de twinning” phenomenon. Combined with the shear strain diagram (Figure 10c_1_), the strain is more severe here. Some twins are impacted by the movement of dislocations at the grain boundary, and dislocations accumulate and penetrate the twin boundary, ultimately resulting in changes in twinning spacing or twinning structure annihilation. In addition, due to the movement of dislocations, twinning originally existing at the grain boundary II later produces an ISF structure. Finally, the grain is compressed and meets the twin boundaries. For the displacement vector diagram of this process, most of the displacement vector values at grain boundaries and within grain II are high, indicating that grain boundaries and twin boundaries are hindering dislocation movement while also undergoing migration and variation. Figure 11 shows the atomic diagram of the compression process of a nanocrystalline polycrystalline AgPd alloy with twin spacing of 6.74 nm and an average grain size of 11.00 nm. After the stress reaches the elastic limit, the two Shockley partial dislocations at the grain boundary and inside of the grain act on the same twin, causing it to eventually divide into several intermittent twin and layer fault structures. In the later stage of deformation, some new twin structures and V-shaped secondary twin structures were generated, as shown in Figure 11d. Here, dislocations bypass TBs via cross-slip, forming V-shaped secondary twins (Figure 11d, black dashed box). Strain localizes at GBs rather than TBs, mimicking twin-free behavior (Figure 8). The outcome is a lower dislocation density but higher ductility due to delayed fracture. At λ = 6.74 nm (>λc), TBs are sparse and act as mild obstacles. Dislocations bypass TBs via cross-slip, forming secondary twins (Figure 11d) and reducing strain hardening. When considering temperature, at low temperatures (10–100 K), dislocation mobility decreases, favoring planar slip and secondary twinning (Figure 11d). At high temperatures (300–500 K), enhanced dislocation climb and TB migration further reduce strain hardening.

The deformation mechanisms in nanotwinned AgPd alloys exhibit a strong dependence on twin spacing. For λ = 0.67 nm, detwinning dominates due to excessive TB density, leading to premature softening (Figure 9). At the critical spacing λ = 1.35 nm, TBs optimally hinder the dislocation motion while resisting detwinning, resulting in peak flow stress (Figure 10). In contrast, larger spacings (λ = 6.74 nm) permit dislocation bypass and secondary twinning, mimicking the plasticity of twin-free systems (Figure 11). These trends align with Wei’s scaling law [8], confirming that λc ≈ 1.35 nm represents the transition from TB-mediated to dislocation-dominated deformation. Low temperatures (10–100 K): small λ (0.67 nm); detwinning is suppressed; and TBs stabilize HCP atoms, enhancing strength (Figure 11a). Large λ (6.74 nm): dislocation mobility decreases, favoring planar slip and secondary twinning (Figure 11d).

It is important to note that the observed deformation mechanisms, such as detwinning and strain localization, are analyzed under the assumption of uniform chemical distribution. In practical AgPd systems, surface segregation—driven by atomic size mismatch and differences in surface energy—could modify grain boundary cohesion and dislocation nucleation sites. For instance, Pd-rich regions at boundaries might enhance local hardening, whereas Ag segregation could promote interfacial sliding. Experimental characterization (e.g., via atom probe tomography) combined with segregation-aware simulations would help quantify these effects.

### 3.3. Effect of Temperature on the Mechanical Behavior of the Nanotwin AgPd Alloy

Some researchers believe that as the temperature of the system increases, the energy obtained by atoms will increase, leading to an increase in their vibration frequency and amplitude, making it easier to move across obstacles, such as grain boundaries, phase boundaries, and solute clusters. Therefore, as the temperature increases, the movement of dislocations and grain boundaries, as well as the diffusion of atoms, become easier. In order to study the effect of temperature on the deformation behavior of nanocrystalline polycrystalline AgPd alloys with different twin spacings, referring to the deformation characteristics of different sizes of configurations in Table 1 and Figure 2, a configuration with an average grain size of 11.00 nm was selected.

Figure 12 presents the stress–strain curves of the nanotwin AgPd alloy under different temperatures. Key trends include a systematic increase in yield strength and Young’s modulus with decreasing temperature. The inverse relationship between temperature and mechanical properties is evident in Figure 13. For instance, at λ = 1.35 nm, Young’s modulus decreases from 71.70 ± 1.2 GPa at 10 K to 49.49 ± 0.9 GPa at 500 K, while the average flow stress declines from 1.81 ± 0.07 GPa to 1.30 ± 0.05 GPa. A paired t-test confirmed the significance of these trends (*p* < 0.01 for all twin spacings). The reduction in thermal energy at cryogenic conditions stabilizes twin boundaries and suppresses dislocation mobility, thereby enhancing both the modulus and the strength. As presented in Table 3, we can observe the specific values of Young’s modulus at different temperatures and twin spacings. For instance, when the temperature is 10 K and the twin spacing is 0.67 nm, the Young’s modulus is 65.25 GPa; at 100 K with the same twin spacing, it is 61.00 GPa. These data clearly illustrate the influence of temperature on the Young’s modulus. At 300 K, the peak flow stress for λ = 1.35 nm is 1.48 ± 0.06 GPa. The turning point (λc) was determined using the derivative of the fitted curve. Uncertainty in λc was calculated via bootstrapping (1000 iterations), yielding λc = 1.35 ± 0.12 nm at 300 K. As the twin spacing decreases, the average flow stress at any temperature shows a trend of first increasing and then decreasing. However, the twinning spacing corresponding to the turning point is different, with 1.35 nm for 100 K and 500 K and 2.02 nm and 3.37 nm for 300 K and 10 K, respectively. Thermal stability: The 4.8% atomic mismatch stabilizes TBs against thermal migration. At 500 K, AgPd retains 70% of its room-temperature strength (Table 3), outperforming pure Pd, which exhibits severe detwinning above 400 K. Functional–mechanical synergy: AgPd’s inherent corrosion resistance and high electrical conductivity (inherited from Ag) are maintained, along with enhanced mechanical properties. This makes it suitable for applications like hydrogen membranes and aerospace components, where monometallic systems face trade-offs.

### 3.4. Effect of Temperature on the Deformation Mechanism of the Nanotwin AgPd Alloy

Figure 14 shows a comparative analysis of the atomic proportions of various structures in AgPd alloys with twin spacing of 1.34 nm after relaxation at different temperatures. As the relaxation temperature increases, the number of FCC and HCP structural atoms decreases, but the number of HCP structural atoms changes less. The number of atoms in the other structure increases accordingly. After the temperature increases, the energy obtained by some atoms in the configuration increases, leading to a transition from FCC and HCP structured atoms to other structured atoms, resulting in an increase in the proportion of disordered structured atoms in the configuration. Figure 15 shows the trend of atomic changes and stress with strain during the deformation process of the AgPd alloy with twin spacing of 1.34 nm at temperatures of 10 K, 100 K, 300 K, and 500 K. When the deformation temperature is low, as shown in Figure 15a 10 K and Figure 15b 100 K, in the initial stage of strain, the FCC structural atoms begin to gradually decrease, while the other structural atoms begin to increase. At this time, the HCP structural atoms remain basically unchanged. When the stress value exceeds the elastic limit and the yield upper limit, the number of HCP structural atoms begins to increase. This process is mainly due to the transition of FCC structural atoms to HCP and other structural atoms, indicating that the stress value is relatively small in the early deformation stage and the HCP structural atoms are relatively stable in the configuration. As the stress value increases in the later stage, the strain increases, and the stress concentration in the grains is more severe. The HCP structural atoms in twinning and stacking fault structures begin to change. When the deformation temperature is high, as shown in Figure 15d 500 K, in the initial stage of deformation, the number of various structural atoms remains basically unchanged. When the stress value increases to the elastic limit, the number of HCP and FCC structural atoms begins to decrease, and the number of other structural atoms begins to increase, indicating that there is a transition from HCP and FCC structural atoms to other structural atoms at this stage. It can be seen that the decrease in temperature to some extent increases the stability of the twin structure.

Figure 16 shows the variation trend of the length of various types of dislocation lines with strain in AgPd alloy with twin spacing of 1.34 nm at different deformation temperatures. During the entire deformation process, Shockley dislocations accounted for the vast majority, followed by Perfect dislocations, while Stair rod, Hirth, and Frank dislocations accounted for very little, almost zero. The length of various dislocation lines after relaxation at 10 K, 100 K, and 300 K is basically not affected by temperature, and there is not much difference between them; However, after relaxation at 500 K temperature, the length of various dislocation lines is less compared to other temperatures, indicating that the extension of dislocations is less after temperature increase. Based on the trend chart of the total dislocation density of the alloy with strain at different temperatures, as shown in Figure 17, during the entire deformation process, increasing the deformation temperature results in a decrease in the corresponding total dislocation density, indicating that temperature has a certain inhibitory effect on the generation of defects such as dislocations and stacking faults.

In order to better compare the influence mechanism of temperature on the nanotwin AgPd alloy, combined with the characteristics of the configuration’s stress–strain curve and the CNA structure number change curve, four stages were selected: relaxation without deformation, strain reaching the elastic limit, FCC structure atom reduction to a stable stage, and deformation completion. The evolution characteristics of dislocations and microstructures with twin spacing of the 1.42 nm configuration at 10 K, 300 K, and 500 K temperatures were analyzed, as shown in Figure 18. When the temperature is low (T = 10 K), after relaxation, some HCP and BCC structural atoms appear at the grain boundaries, with a higher proportion compared to other temperatures. Deformation reaches the elastic limit, and twinning dislocations sprout at the black dashed line of grain I and move in the direction of the arrow in the figure, causing the twinning to move away from adjacent twinning towards the grain boundary. As deformation continues, twinning in grains II and IV is severely damaged, resulting in the formation of ISF and ESF structures. It is worth noting that the transformation process from the ESF to the ISF structure has been observed, with some exhibiting an ESF structure and the others exhibiting an ISF structure, and new twinning appears at the junction of these two grains. Although the twinning length here is not long due to the obstruction effect of grain boundaries, it alleviates the degree of stress concentration here. After deformation to 15%, the black solid line of grain III has transformed into a shorter ISF structure, and new twinning has appeared at the grain boundary. When the temperature is high (T = 500 K), after relaxation is completed, a certain amount of FCC and HCP structural atoms appear between the grain boundaries and within the grains, transforming into other disordered structural atoms. These defects provide nucleation points for dislocation germination to a certain extent. Compared to low temperatures, the small grains at the grain boundaries marked by the yellow circle temporarily disappear during the relaxation process and gradually appear again during subsequent deformation, indicating that higher temperatures promote the activity of grain boundaries. Deformation reaches the elastic limit, and the transformation of disordered structures within the grain increases. Dislocation sources appear at the twin boundaries within grain II and further move towards the grain boundaries, ultimately leading to a decrease in twin spacing and partial twinning annihilation. When the deformation reaches 10.00%, the number of atoms in the FCC structure enters a slow fluctuation stage. At this stage, ESF structures appear near the grain boundaries of grain II, and the twinning spacing within other grains decreases to varying degrees. After deformation, the grain boundaries within grains II and IV were severely damaged, and twinning within grain II almost disappeared. Large disordered atomic clusters appeared within grain IV.

In order to determine the degree of influence of temperature factors on the average flow stress of the AgPd alloy with twin spacing of 0.67 nm at a room temperature of 300 K, which is lower than that of other twin spacings, the deformation process of this twin spacing was selected at temperatures of 10 K and 500 K. Figure 19 shows the atomic diagram of the nanocrystalline polycrystalline AgPd alloy with twin spacing of 0.67 nm and an average grain size of 11.00 nm during 10 K compression. After relaxation, the twin structure and the intra crystal atoms did not change. After strain reached the elastic limit, the phenomenon of “de twinning” appeared in grain I. The reason was that the dislocations emitted at the grain boundary met with the twin here, forcing the twin boundary to migrate towards the direction of dislocation movement, as shown in Figure 19b) When the strain reached 13.05%, the twin structure inside of grain III was basically annihilated and transformed into structures like ISF and ESF. After the strain was completed, the ISF structure shown in the black box also disappeared. For AgPd alloys with the same parameters compressed at a temperature of 500 K, as shown in Figure 20, after relaxation, some atoms in the twin boundaries and grains transform into disordered structural atoms, providing a location for subsequent dislocation source emission. After reaching the elastic limit, the phenomenon of “de twinning” occurs in grain III. In the later stage of deformation, the twin structure also transforms into an ESF structure, resulting in a decrease in the number of twin structures. By analyzing the changes in the density of various structural atoms and dislocations at two different temperatures, it was found that there was not much difference in the reduction of HCP structural atoms. However, at 500 K, the main transition occurred from HCP structural atoms to disordered structural atoms, while at 10 K the transition occurred to FCC and disordered structural atoms. Higher temperatures to some extent intensified the trend of atomic motion. This exacerbates the annihilation of twin structures.

### 3.5. Real-World Applications

The findings of this study have significant implications for various real-world applications of AgPd alloys, particularly in industries where mechanical strength, thermal stability, and functional performance are crucial. For hydrogen storage and catalysis, optimizing twin spacing can enhance the material’s performance by improving mechanical stability during cyclic hydrogenation, thus reducing the risks of embrittlement. Additionally, cryogenic cooling during fabrication can stabilize nanotwins, benefiting long-term durability in hydrogen-rich environments.

In aerospace and high-temperature sensor applications, AgPd alloys are valued for their oxidation resistance and conductivity. The study suggests that larger twin spacings (λ > 6.74 nm) improve ductility and fracture resistance at elevated temperatures, which is essential for components exposed to thermal cycling. Furthermore, understanding the detwinning kinetics is crucial for predicting material degradation in such applications.

For biomedical implants and microelectronics, the alloy’s intermediate twin spacing (λ ≈ 1.35–3.37 nm) can maximize fatigue resistance, which is vital for load-bearing implants. Nanocrystalline structures can suppress detwinning, maintaining surface integrity and reducing wear in electronic contacts, contributing to enhanced performance in biomedical and electronic applications.

## 4. Conclusions

This paper simulates the compression deformation process of nanocrystalline polycrystalline AgPd alloys with different twin spacings using molecular dynamics methods and explores the effects of twin spacing, deformation temperature, and strain rate on the mechanical properties and deformation mechanism of AgPd alloys during deformation. The results show the following.

(1) The average flow stress of the nanotwin AgPd alloys with average grain sizes of 5.50 nm, 8.24 nm, and 11.00 nm increases first and then decreases with the decrease in twinning spacing. Among them, the alloy with an average grain size of 11.00 nm increases and then decreases with a decrease in twinning spacing of λ = 0.67. The average flow stress at 1.35 nm reaches its maximum value. The average flow stress of the nanotwin AgPd alloy with an average grain size of 13.76 nm continued to increase with the decrease in twin spacing, and there was no turning trend. The average flow stress of the nanocrystalline polycrystalline AgPd alloy with an added twin structure is higher than that without the twin structure. During the deformation process of a nanotwin AgPd alloy with an average grain size of 11.00 nm, the alloy with twin spacing of 0.67 nm experiences a phenomenon of “de twinning”. The alloy with twin spacing of 6.74 nm exhibits new twin structures and V-shaped secondary twinning.

(2) By changing the relaxation temperature and deformation temperature of the nanotwin AgPd alloy with an average grain size of 11.00 nm, it was found that as the temperature increased, the Young’s modulus and average flow stress of the nanotwin AgPd alloy gradually decreased. On the microstructure, it was found that the AgPd alloy with twin spacing of 1.35 nm transformed from HCP and FCC structural atoms to other structural atoms at higher temperatures (500 K) and from FCC structural atoms to HCP and other structural atoms at lower temperatures (10 K and 100 K). The length and total dislocation density of the Shockley dislocation lines, which occupy the main position in the dislocation lines, increased with decreasing temperature. Lower temperatures will increase the stability of preset twins and the generation of new twins, while higher temperatures will promote the movement of atoms and the migration of grain boundaries and twin boundaries, further exacerbating the occurrence of “de twinning”.

While this study elucidates the interplay between twin spacing, temperature, and deformation mechanisms in nanotwinned AgPd alloys, the omission of surface segregation effects represents a notable limitation. Surface-driven chemical heterogeneity, inherent to bimetallic systems, may significantly influence dislocation dynamics and twin stability. Extending the model to incorporate segregation thermodynamics and kinetics, alongside experimental validation, will be critical for advancing the design of next-generation nanotwinned alloys.

## Figures and Tables

**Figure 1 nanomaterials-15-00323-f001:**
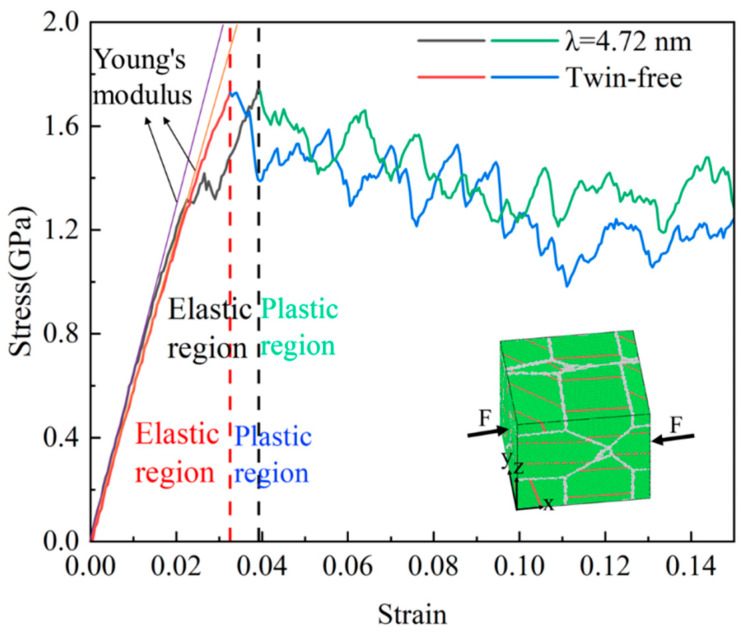
Stress–strain curve of silver palladium nanoalloy with an average grain size of 11.00 nm.

**Figure 2 nanomaterials-15-00323-f002:**
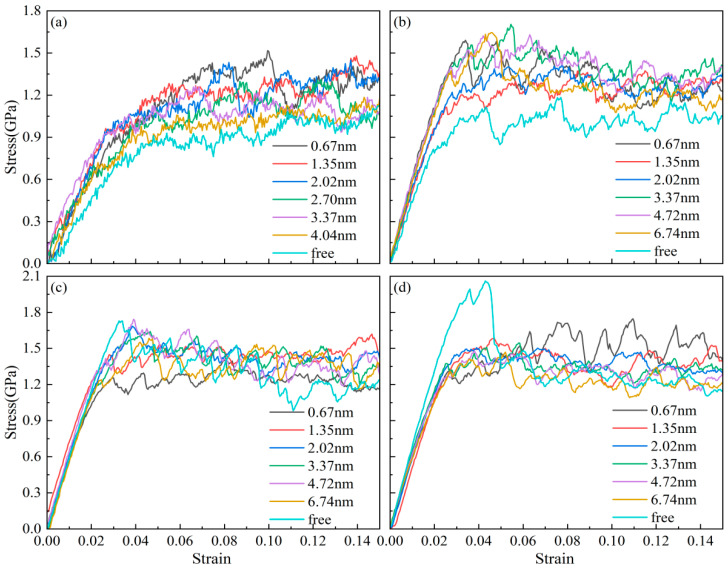
The stress–strain curves of the nanotwin AgPd alloy with different twin spacings under compression. (**a**) d = 5.50 nm; (**b**) d = 8.24 nm; (**c**) d = 11.00 nm; (**d**) d = 13.76 nm.

**Figure 3 nanomaterials-15-00323-f003:**
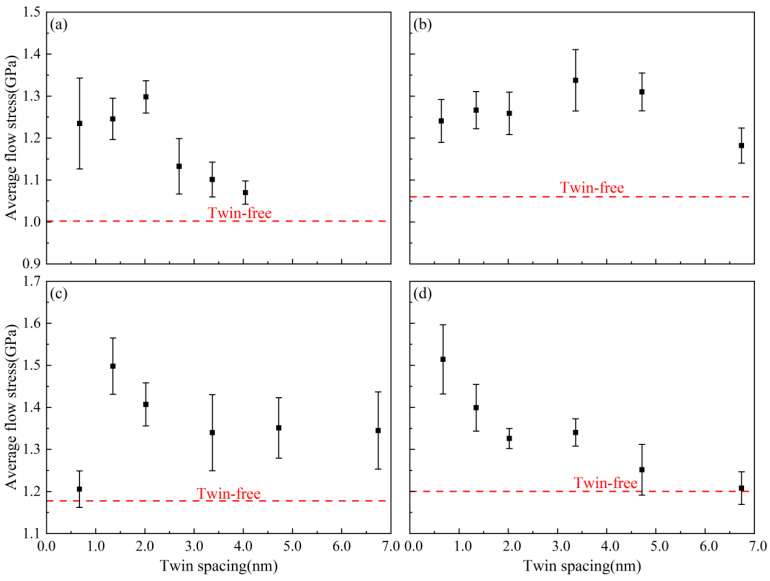
The average flow stress variation curve of the nanotwin AgPd alloy with different twin spacings during compression. (**a**) d = 5.50 nm; (**b**) d = 8.24 nm; (**c**) d = 11.00 nm; (**d**) d = 13.76 nm.

**Figure 4 nanomaterials-15-00323-f004:**
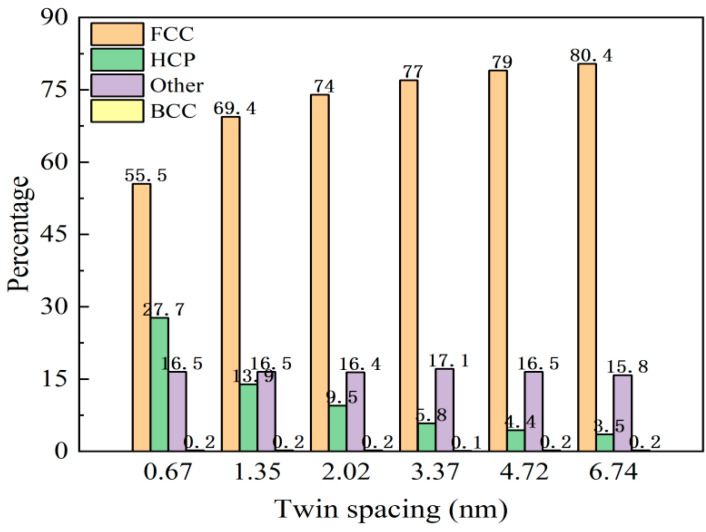
Atomic proportions of FCC, HCP, and disordered structures in the nanotwin AgPd alloy (average grain size d = 11.00 nm) after relaxation under different twin spacings (λ = 0.67–6.74 nm).

**Figure 5 nanomaterials-15-00323-f005:**
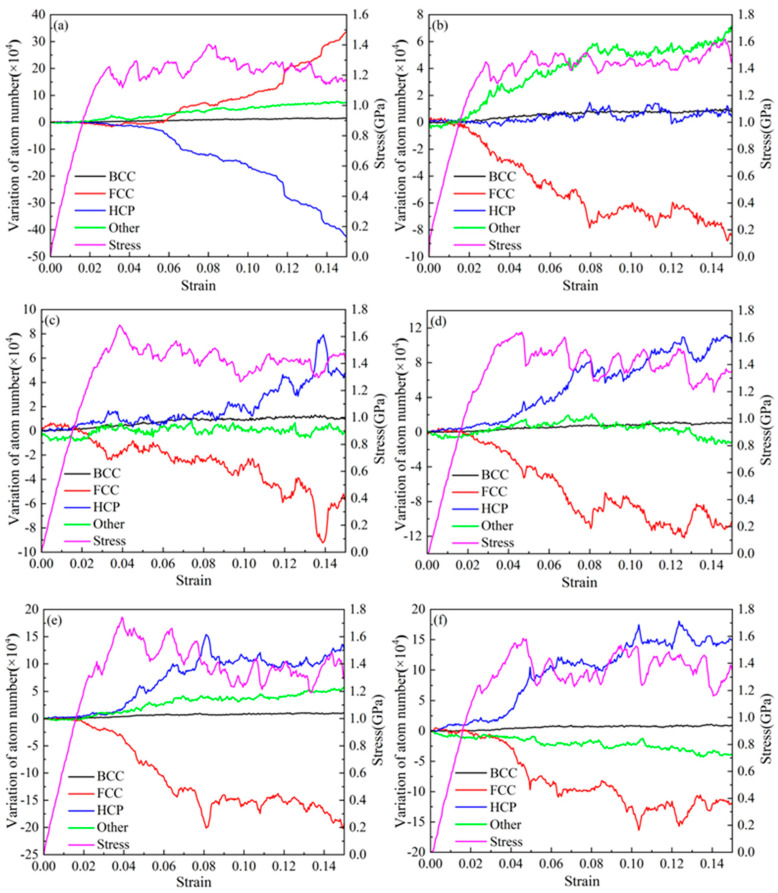
Variation of atomic structure proportions (FCC, HCP, other) and stress with strain for nanotwin AgPd alloy (average grain size d = 11.00 nm) under different twin spacings (λ = 0.67–6.74 nm): (**a**) λ = 0.67 nm; (**b**) λ = 1.35 nm; (**c**) λ = 2.02 nm; (**d**) λ = 3.37 nm; (**e**) λ = 4.72 nm; (**f**) λ = 6.74 nm.

**Figure 6 nanomaterials-15-00323-f006:**
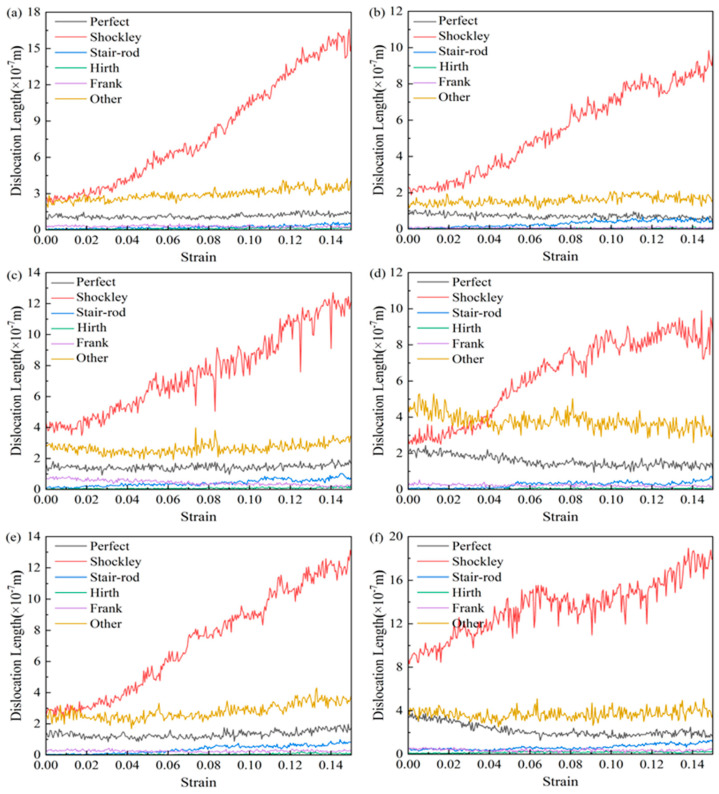
Evolution of dislocation line lengths (Shockley, Perfect, Stair rod) with strain in the nanotwin AgPd alloy (average grain size d = 11.00 nm) for twin spacings λ = 0.67–6.74 nm. (**a**) λ = 0.67 nm; (**b**) λ = 1.35 nm; (**c**) λ = 2.02 nm; (**d**) λ = 3.37 nm; (**e**) λ = 4.72 nm; (**f**) λ = 6.74 nm.

**Figure 7 nanomaterials-15-00323-f007:**
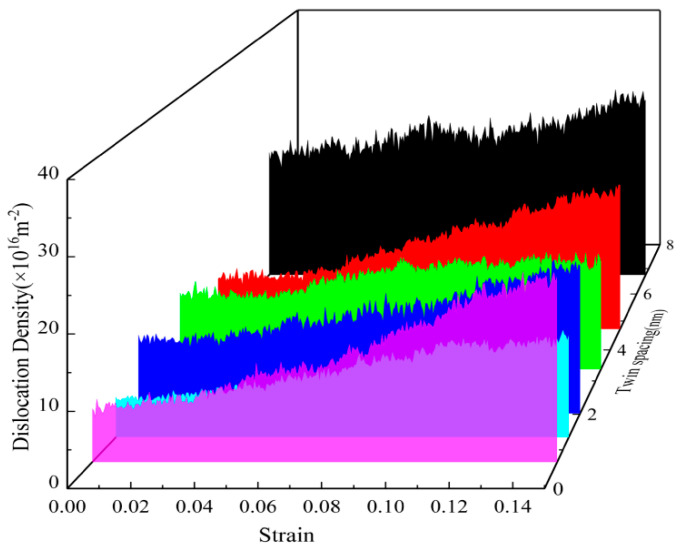
Dislocation density evolution with strain in the nanotwin AgPd alloy (average grain size d = 11.00 nm) under varying twin spacings (λ = 0.67–6.74 nm).

**Figure 8 nanomaterials-15-00323-f008:**
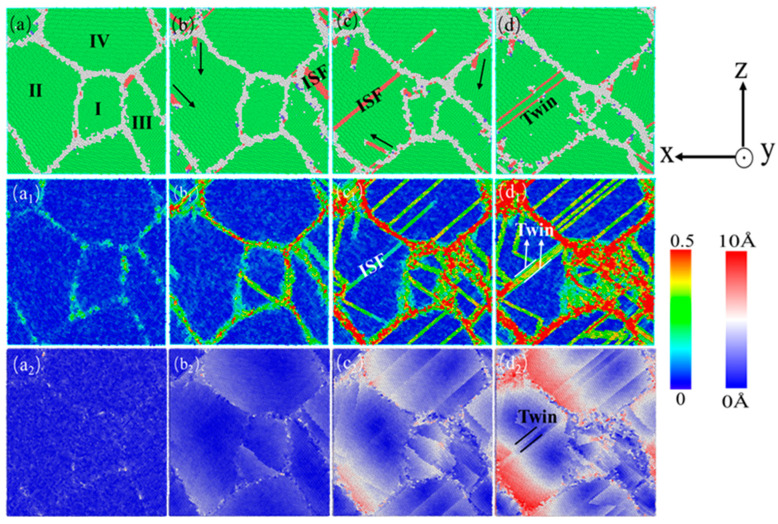
The nanocrystalline polycrystalline AgPd alloy with an average grain size of 11.00 nm exhibits compressed atomic diagrams at 0% (**a**), 3.00% (**b**), 12.00% (**c**), and 15.00% (**d**). (**a_1_**–**d_1_**) and (**a_2_**–**d_2_**) are the corresponding shear strain and displacement vector diagrams, respectively.

**Figure 9 nanomaterials-15-00323-f009:**
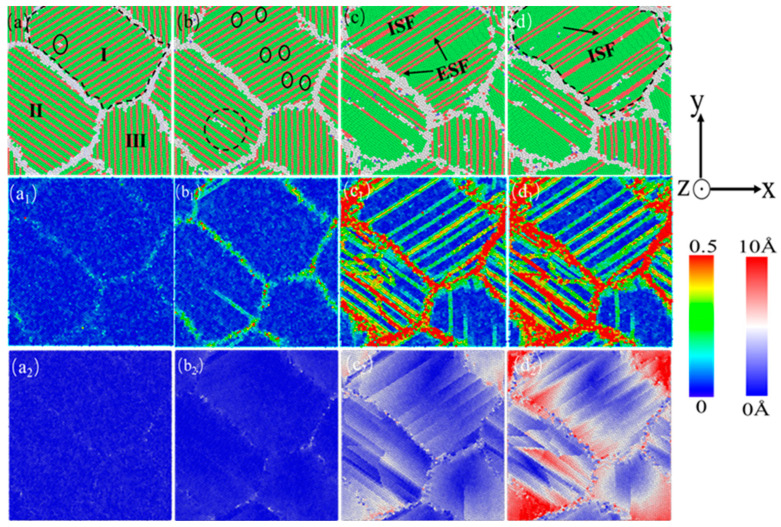
The nanocrystalline polycrystalline AgPd alloy with twin spacing of 0.67 nm and an average grain size of 11.00 nm exhibits compressed atomic diagrams at 0% (**a**), 3.00% (**b**), 12.00% (**c**), and 15.00% (**d**). (**a_1_**–**d_1_**) and (**a_2_**–**d_2_**) are the corresponding shear strain and displacement vector diagrams, respectively.

**Figure 10 nanomaterials-15-00323-f010:**
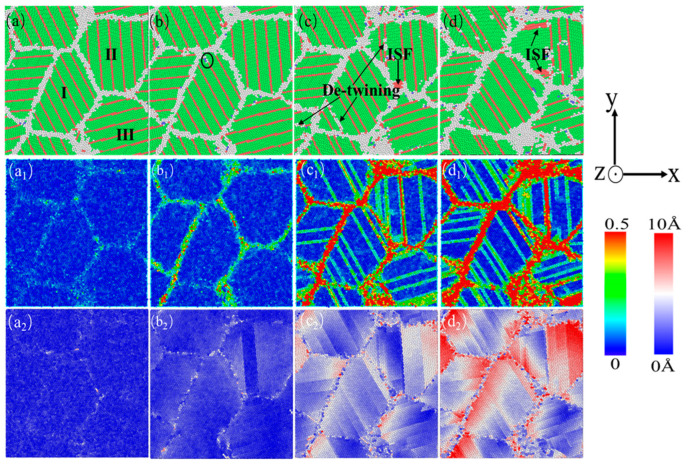
The nanocrystalline polycrystalline AgPd alloy with twin spacing of 1.34 nm and an average grain size of 11.00 nm exhibits compressed atomic diagrams at 0% (**a**), 2.85% (**b**), 8.30% (**c**), and 15.00% (**d**). (**a_1_**–**d_1_**) and (**a_2_**–**d_2_**) are the corresponding shear strain and displacement vector diagrams, respectively.

**Figure 11 nanomaterials-15-00323-f011:**
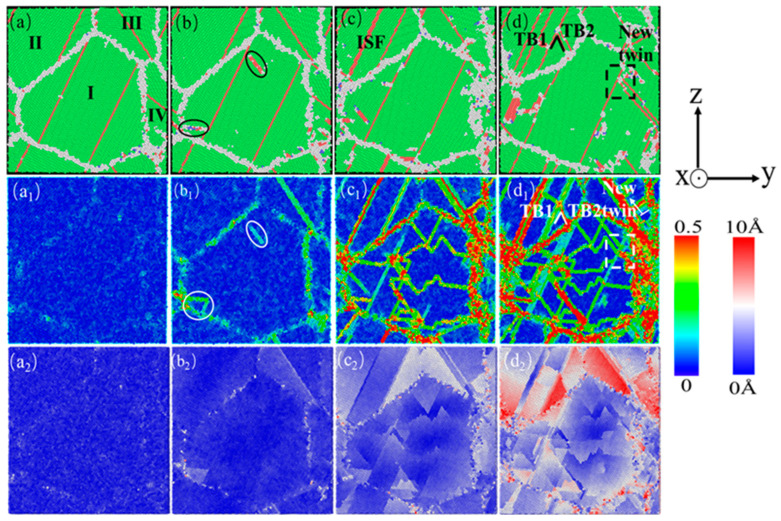
The nanocrystalline polycrystalline AgPd alloy with twin spacing of 6.74 nm and an average grain size of 11.00 nm exhibits compressed atomic diagrams at 0% (**a**), 2.45% (**b**), 8.55% (**c**), and 15.00% (**d**). (**a_1_**–**d_1_**) and (**a_2_**–**d_2_**) are the corresponding shear strain and displacement vector diagrams, respectively.

**Figure 12 nanomaterials-15-00323-f012:**
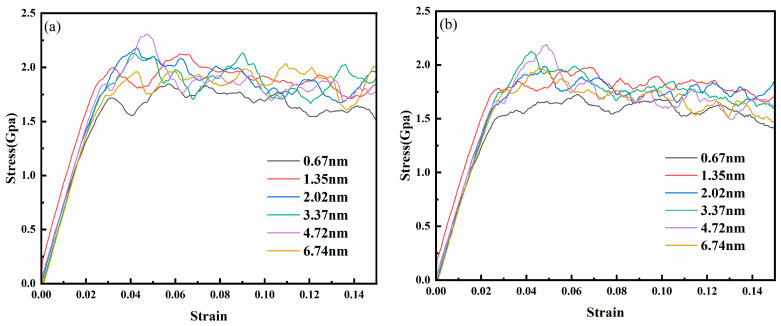
The stress–strain curves of the nanotwin AgPd alloy under compression at different temperatures: (**a**) 10 K; (**b**) 100 K; (**c**) 300 K; (**d**) 500 K.

**Figure 13 nanomaterials-15-00323-f013:**
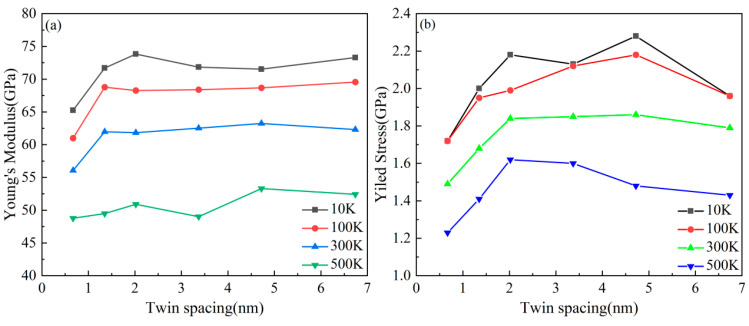
The curve of the change in Young’s modulus (**a**) and the average flow stress (**b**) of the nanotwin AgPd alloy at different temperatures with variation in nanotwin spacing.

**Figure 14 nanomaterials-15-00323-f014:**
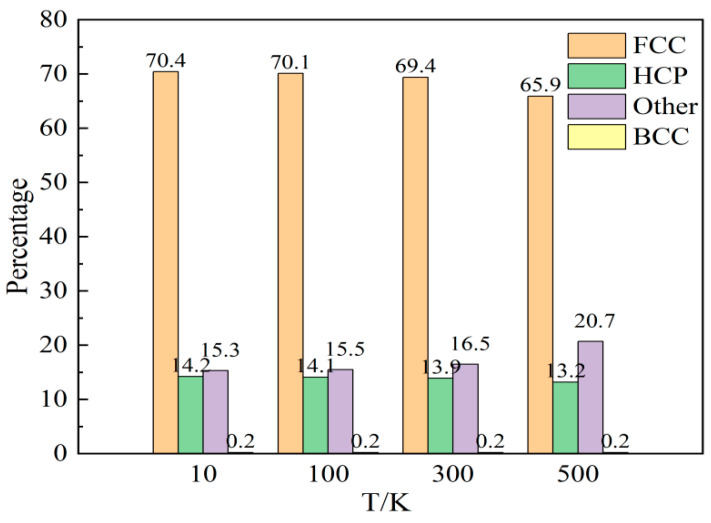
Statistical charts of the atomic proportions of each structure in the nanotwin AgPd alloy after relaxation at different temperatures.

**Figure 15 nanomaterials-15-00323-f015:**
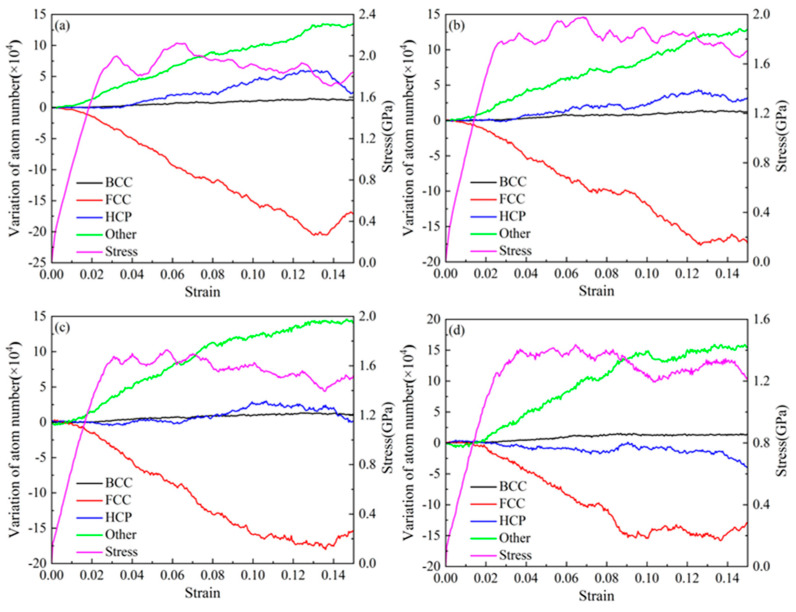
The variation curves of atomic changes and stress with strain in the structure of the nanotwin AgPd alloy at different temperatures. (**a**) 10 K; (**b**) 100 K; (**c**) 300 K; (**d**) 500 K.

**Figure 16 nanomaterials-15-00323-f016:**
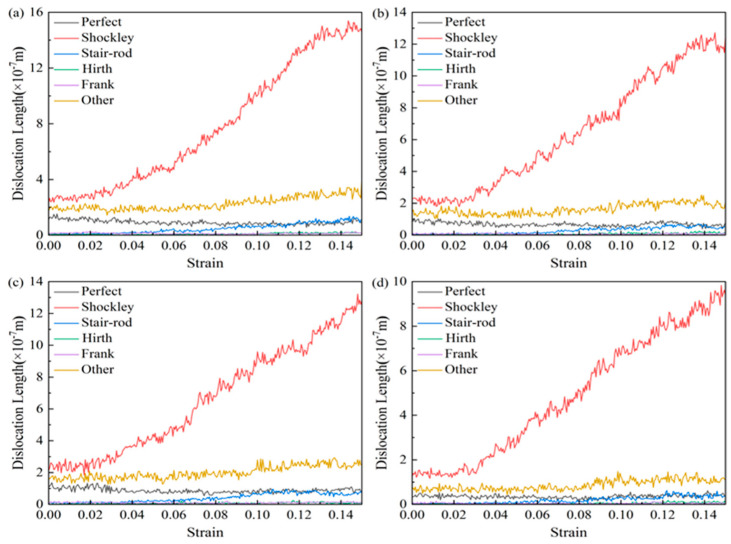
The variation curve of the length of various types of dislocation lines with strain in the nanotwin AgPd alloy at different temperatures. (**a**) 10 K; (**b**) 100 K; (**c**) 300 K; (**d**) 500 K.

**Figure 17 nanomaterials-15-00323-f017:**
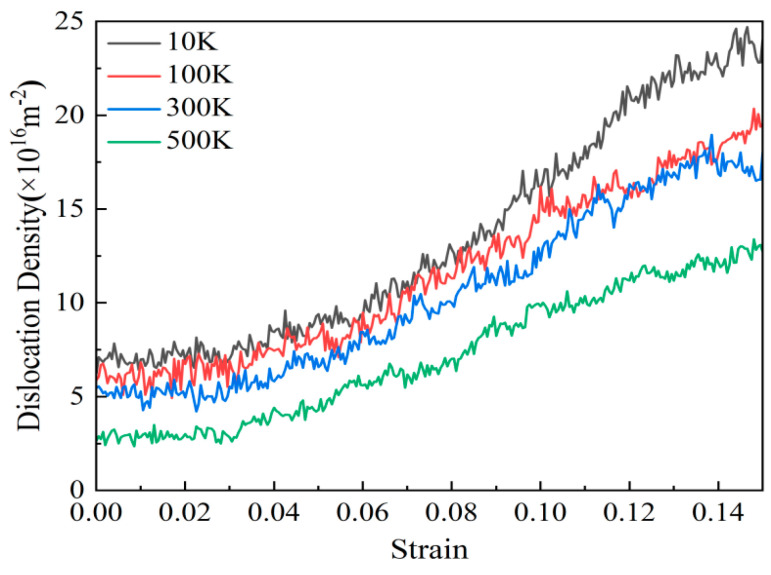
The variation curve of dislocation density with strain in the nanotwin AgPd alloy at different temperatures.

**Figure 18 nanomaterials-15-00323-f018:**
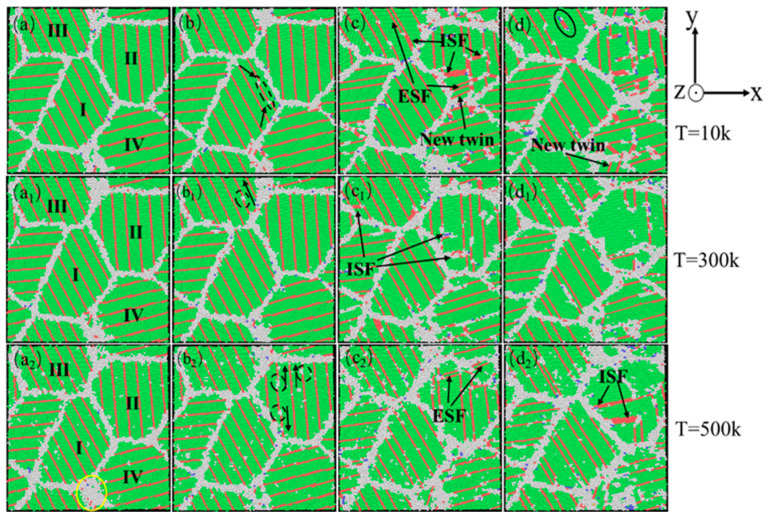
The compressed atomic diagrams of the nanotwin AgPd alloy at different temperatures at strains of 0% (**a**,**a_1_**,**a_2_**), 2.60% (**b**,**b_1_**,**b_2_**), 10.00% (**c**,**c_1_**,**c_2_**), and 15.00% (**d**,**d_1_**,**d_2_**).

**Figure 19 nanomaterials-15-00323-f019:**
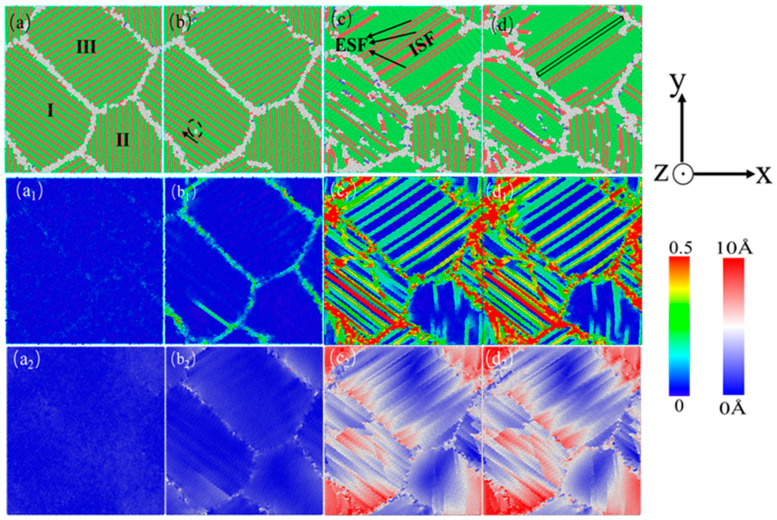
(**a**–**d**) The compressed atomic diagrams of a nanocrystalline polycrystalline AgPd alloy with twin spacing of 0.67 nm and an average grain size of 11.00 nm at 0%, 3.15%, 13.05%, and 15.00% strains at 10 K are presented. (**a_1_**–**d_1_**) and (**a_2_**–**d_2_**) are the corresponding shear strain diagrams and displacement vector diagrams, respectively.

**Figure 20 nanomaterials-15-00323-f020:**
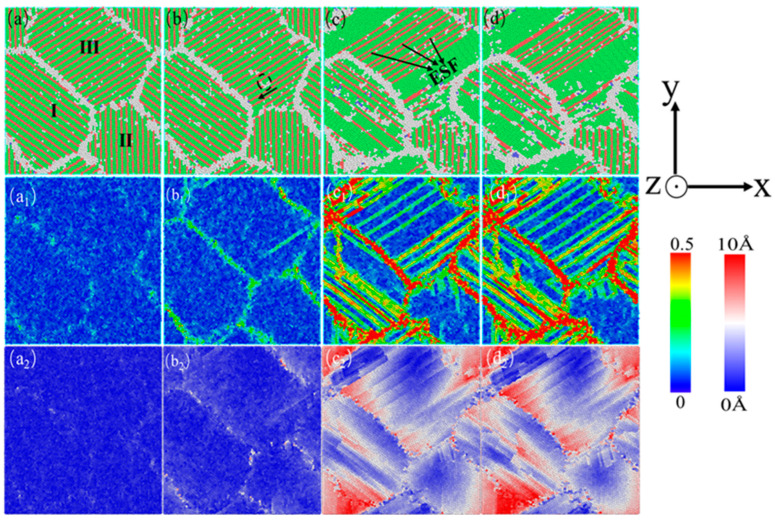
(**a**–**d**) The compressed atomic diagrams of a nanocrystalline polycrystalline AgPd alloy with twin spacing of 0.67 nm and an average grain size of 11.00 nm at 0%, 3.15%, 13.05%, and 15.00% strains at 500 K are presented. (**a_1_**–**d_1_**) and (**a_2_**–**d_2_**) are the corresponding shear strain diagrams and displacement vector diagrams, respectively.

**Table 1 nanomaterials-15-00323-t001:** Initial configuration-related parameters of the nanotwin AgPd alloy.

Alloy	Average Grain Size (d/nm)	Number of Atoms	Twin Spacing (λ/nm)					
AgPd	5.00	67,900	0.67	1.35	2.02	2.70	3.37	4.04
	8.24	228,500	0.67	1.35	2.02	3.37	4.72	6.74
	11.00	542,500	0.67	1.35	2.02	3.37	4.72	6.74
	13.76	1,061,000	0.67	1.35	2.02	3.37	4.72	6.74

**Table 2 nanomaterials-15-00323-t002:** Statistical analysis of mechanical properties of the nanotwin AgPd alloy with average grain sizes of 11.00 nm and 13.76 nm.

Average Grain Size (d/nm)	Twin Spacing (λ/nm)	Young’s Modulus (GPa)	Yield Strength (GPa)	Average Flow Stress (GPa)	Elastic–Plastic Strain Transition Value
11.00	0.67	56.07	1.49	1.21	0.029
	1.35	61.97	1.68	1.50	0.028
	2.02	61.84	1.84	1.41	0.038
	3.37	62.52	1.85	1.34	0.041
	4.72	63.24	1.86	1.35	0.039
	6.74	62.31	1.79	1.34	0.040
	twin-free	62.52	2.02	1.17	0.033
13.76	0.67	54.22	1.45	1.51	0.041
	1.35	61.06	1.57	1.39	0.047
	2.02	61.00	1.50	1.33	0.040
	3.37	58.08	1.54	1.34	0.043
	4.72	57.24	1.50	1.25	0.041
	6.74	61.46	1.49	1.21	0.039
	twin-free	67.62	2.06	1.20	0.036

**Table 3 nanomaterials-15-00323-t003:** Statistical analysis of mechanical properties related to deformation of the nanotwin AgPd alloy with an average grain size of 11.00 nm at different temperatures.

Deformation Temperature (T/K)	Twin Spacing (λ/nm)	Young’s Modulus (GPa)	Yield Strength (GPa)	Average Flow Stress (GPa)	Elastic–Plastic Strain Transition Value
10	0.67	65.25	1.72	1.59	0.031
	1.35	71.70	2.00	1.81	0.034
	2.02	73.83	2.18	1.80	0.042
	3.37	71.83	2.13	1.86	0.041
	4.72	71.53	2.28	1.80	0.045
	6.74	73.29	1.96	1.82	0.043
100	0.67	61.00	1.72	1.54	0.038
	1.35	68.78	1.95	1.75	0.039
	2.02	68.27	1.99	1.76	0.047
	3.37	68.39	2.12	1.66	0.042
	4.72	68.68	2.18	1.62	0.048
	6.74	69,356	1.96	1.58	0.045
300	0.67	61.00	1.49	1.27	0.033
	1.35	61.97	1.68	1.48	0.031
	2.02	61.84	1.84	1.55	0.041
	3.37	62.52	1.85	1.52	0.042
	4.72	63.24	1.86	1.44	0.042
	6.74	62.31	1.79	1.38	0.048
500	0.67	48.78	1.23	1.11	0.035
	1.35	49.49	1.41	1.30	0.036
	2.02	50.91	1.62	1.30	0.045
	3.37	49.02	1.60	1.24	0.045
	4.72	53.31	1.48	1.21	0.043
	6.74	52.42	1.43	1.24	0.042

## Data Availability

Data are contained within the article.

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
