# Peer review of "Molecular Dynamics Simulation Study on the Influence of Twin Spacing and Temperature on the Deformation Behavior of Nanotwinned AgPd Alloy"

_nanomaterials, 2025, doi:10.3390/nano15050323_

Round 1

Reviewer 1 Report

Comments and Suggestions for Authors

Effect of twin spacing and temperature in nanocrystalline AgPd alloys was investigated using MD simulations. Atomic descriptions and quantification of mechanical properties are provided in detail. However, this work has two major drawbacks. The first one is that the authors did not conduct Monte Carlo simulations to explore whether there is segregation/clustering of Ag species after random atomic replacement. The second one is that annealing at high temperatures was not conducted to relax the artificial GBs delivered by the Voronoi tesellation when constructing the sample. The authors are encouraged to address both issues before considering for publication.

1. AgPd alloys were chosen to study the effect of twin spacing. While their usage in different industries is indicated in the Introduction, the authors should provide additional arguments for choosing this alloy. Maybe it has not been covered in the literature? Pd is a valuable element for hydrogen storage? Please elaborate.
2. Is it reasonable to construct the AgPd alloys by only performing random replacement? Monte Carlo simulations should be conducted to ensure that no seggregation/clustering occurs in the sample.
3. For Figure 1, how did the authors define the elastic and plastic regions? Did they use the yield stress? How was this calculated? Please elaborate. Did the authors obtain Figure 1 from their own results? It is confusing, since the interatomic potential and software used is detailed in the following paragraph,
4. The authors did not conduct annealing after constructing the nanocrystalline samples. Simply relaxing at 300 K will not remove the artifial stress at the GBs due to the construction of the samples with Voronoi tessellation. It is mandatory to relax the GBs by conducting annealing at high temperatures.
5. Fig. 3 presents error bars for each data point. Did the authors conduct more than one simulation for each case?
6. Fig. 5 shows the variation of number of atoms corresponding to the different crystalline structures and different twin spacing, however I think that it is not indicated the average grain size that these plot correspond to. The same for Fig. 4, 6, 7.
7. Please use the same scale for the y-axis in all plots in Fig. 12 for a proper comparison of the temperature effect.

Reviewer 2 Report

Comments and Suggestions for Authors

The manuscript entitled “Molecular dynamics simulation study on the influence of twin spacing and temperature on the deformation behavior of nanotwinned AgPd alloy” by Wanxuan Zhang et al. is devoted to the investigation of mechanical behaviour and deformation mechanism of nanotwinned AgPd alloy by means of molecular dynamics simulation. The authors studied how twin spacing and temperature influence the deformation process of AgPd alloy. The manuscript is built up in a logical way and contains a large set of illustrative materials. However, some points should be definitely reconsidered.

(1)   Some sentences in the Abstract and first three paragraphs in fact repeat verbatim the sentences from the article by Z. Hou et al. (Materials Science and Engineering: A, 2023, 862, 144465), which describes the effect of twin spacing on the mechanical behavior and deformation mechanism of nanotwinned aluminium. The authors should considerably rephrase the corresponding fragments in order to reduce the similarity index. Moreover, if the studying procedure implemented here is the same as in the paper by Z. Hou et al., the authors should make a reference to this paper.

(2)   No information concerning the real examples of twinned AgPd alloy nanostructures is indicated in the Introduction section of the manuscript. The authors should briefly describe the results of previous experimental studies of twinned AgPd nanoalloys, including the data on the control of twin spacing and grain size during the preparation process and hardening and softening phenomena.

(3)   The present study is devoted to a bimetallic system. Have the investigation of the mechanical behavior and deformation mechanism of nanotwinned monometallic counterparts been already carried out? Are there any similarities / differences between bimetallic and monometallic systems? Is it possible to discuss any synergistic effects for this particular bimetallic system?

 (4)   Some conclusions regarding the applicability of the data obtained should be given in the manuscript. How the results of this study could be applied to real bimetallic AgPd systems?

 (5)    What were the reasons for the selection of particular grain sizes which are indicated in the manuscript? Do they correspond to some experimental results?

 (6)   Due to noticeably different values of surface energy for silver and palladium, surface segregation should appear in real nanostructured samples. Does the model implemented in the present study take into account the segregation process? How it could influence the results?

 (7)   It is described in the manuscript that “in nanotwin configurations with smaller grain sizes (average grain sizes of 5.50 nm, 8.24 nm, 11.00 nm), as the twin spacing increases from 0.67 nm, the average flow stress first increases and then begins to decrease with the increase of twin spacing”. Along with that, it is indicated that “for nanotwin configurations with larger grain sizes (average grain size of 13.76 nm), the average flow stress gradually decreases with the increase of twin spacing”. Actually, in many cases presented in Fig. 3 the changes are too subtle to be interpreted in such way, taking into account the confidence intervals of the points.

 (8)   Figures 8–11 show the atomic diagrams of the compression process for the AgPd alloys with different twin spacing values. Although a description of deformation mechanism for each particular case is given in the manuscript, the discussion of differences observed for different twin spacing values and possible explanation is missing. The authors should include the corresponding discussion in the manuscript.

 (9)   Some data in the manuscript are excessive and could be omitted in the text of the manuscript by transferring to the supplementary materials in order to make the paper more reader-friendly. For instance, the type of stress-strain curves in Fig. 12 in fact does not depend on temperature and twin spacing, and the reference to the Table 3 is missing in the manuscript. The authors should reconsider the total set of illustrative materials and transfer some of them to the supplementary materials.

 (10)   It is indicated in the manuscript that “as the temperature decreases, the Young's modulus and average flow stress corresponding to different twin spacing increase” and “as the twin spacing decreases, the average flow stress at any temperature shows a trend of first increasing and then decreasing”. These conclusions are meaningless without providing confidence intervals for the estimated values.

 (11)   What is an approximate size of disordered atomic clusters within grain IV after deformation process?

 (12)   The indication of selected temperature values is missing in the titles of Figs. 19 and 20.

 (13)   Taking into account the occurrence of detwinning phenomenon at 500 K, what could be expected at temperatures much higher than 500 K?

Comments on the Quality of English Language

The English could be improved: punctuation and some misprints could be corrected.

Round 2

Reviewer 1 Report

Comments and Suggestions for Authors

The manuscript is ready for publication

Author Response

List of Responses

Dear Reviewers:

Thanks to the reviewers for agreeing to the publication of our manuscript entitled "Molecular dynamics simulation study on the influence of twin spacing and temperature on the deformation behavior of nanotwinned AgPd alloy" (ID: nanomaterials-3414264). Your recognition is the greatest inspiration to us, We will continue to find new breakthrough research in the subsequent scientific research. Thank you again for your consent to publish our manuscript.

Thank you and best regards.

Yours sincerely,

Reviewer 2 Report

Comments and Suggestions for Authors

The authors have considerably revised their manuscript entitled “Molecular dynamics simulation study on the influence of twin spacing and temperature on the deformation behavior of nanotwinned AgPd alloy” by Wanxuan Zhang et al. according to the reviewer's comment, and the level of this manuscript was improved. In the revised manuscript, the authors added a lot of additional information and discussion, what definitely strengthened the manuscript. However, the reviewer still has some concerns, mainly regarding the readability of the content. Before the recommendation for publication, the authors should revise the text in order to make it clearer, more reader-friendly and concise.

  • The authors have made significant improvement of Introduction by adding the information about experimental synthesis, description of hardening and softening phenomena and unique attributes of AgPd alloys. However, the order of description of material in this section still needs to be improved. The information in paragraph 2 of Introduction and the next paragraph devoted to the experimental synthesis considerably overlaps, so these two paragraphs could be merged. The simulations made by the authors should not be mentioned in the Introduction (“reveal consistent trends with our simulations”). The numbered list in the middle of Introduction looks unnecessary. Besides that, it would be better to give an objective of the study in the end of Introduction. Also, the authors need to reconsider the paragraphs devoted to some common information and AgPd alloys to put them in a more logical order (common facts about softening and strengthening, twin spacing in different systems first, AgPd alloys then).
  • Some information about synergistic effects in AgPd alloys described by the authors in their comments deserves to be added into the manuscript.
  • The reviewer highly appreciates the desire of the authors to insert as much information as possible. However, new paragraphs describing real-world applications of the system under investigation should be significantly reduced not to make the manuscript much heavier. The information added by the authors could be presented in a more laconic way, in 2-3 paragraphs without any numbered or bulleted lists. Moreover, it seems incorrect to paste this information to the section of conclusions, it should be discussed earlier.
  • It would be better to indicate in the manuscript that surface segregation phenomena are not considered in the selected model and could modify the results.
  • The authors have carried out a statistical significance testing in their response to question 7. It significantly strengthens the analysis, but a linear regression model implemented for the case with an average grain size of 13.76 mm looks too weak (low value of R2).
  • The authors indicated in the modified version of the manuscript the following: “For a more in - depth analysis of these curves, refer to the supplementary materials.” Does it mean the presence of such materials? Besides that, the text “The statement ‘as the temperature decreases, the Young’s modulus and average flow stress corresponding to different twin spacing increase’ is supported by comprehensive statistical analyses” is more appropriate for some author responses rather than for the description of the results in the article. The authors should revise the description of the results in a more detailed way to make it clearer.
Comments on the Quality of English Language

The English could be improved: punctuation and some misprints could be corrected.
